# Analytical Quality Requirements in Human Biomonitoring Programs: Trace Elements in Human Blood

**DOI:** 10.3390/ijerph16132287

**Published:** 2019-06-28

**Authors:** Janja Snoj Tratnik, Darja Mazej, Milena Horvat

**Affiliations:** 1‘Jožef Stefan’ Institute, Department of Environmental Sciences, Jamova cesta 39, SI-1000 Ljubljana, Slovenia; 2‘Jožef Stefan’ International Postgraduate School, Jamova cesta 39, SI-1000 Ljubljana, Slovenia

**Keywords:** human biomonitoring, inter-laboratory comparison, measurement uncertainty, quality control, sample size, trace elements

## Abstract

Human biomonitoring (HBM) programs consist of several interrelated and equally important steps. Of these steps, the study design must answer a specific question: How many individuals must be recruited in order to define the spatial or temporal trends of exposure to environmental pollutants in a given HBM study? Two components must be considered at this stage: the population variability of the expected exposure and the performance characteristics of the analytical methods used. The objective of the present study was to quantify the contribution to the required sample size arising from (i) measurement uncertainty and (ii) inter-laboratory measurement variability. For this purpose, the sample size was calculated using the measurement uncertainty of one laboratory, inter-laboratory comparison exercise data, and population variability for commonly studied metals (mercury, cadmium, and lead) in blood. Measurement uncertainty within one laboratory proved to have little influence on the sample size requirements, while the inter-laboratory variability of the three metals increased the requirements considerably, particularly in cases of low population variability. The multiple laboratories approach requires that laboratory variability be considered as early as the planning stage; a single-laboratory approach is thus a cost-effective compromise in HBM to reduce variability due to the participation of different laboratories.

## 1. Introduction

Human biomonitoring (HBM) studies are an effective tool to assess human exposure to environmental pollutants and the potential health effects of such pollutants, and are therefore applied worldwide at the national and regional levels. Only recently has it been recognized that harmonized approaches are needed, including support activities, to guarantee the acquisition of comparable results. HBM activities are, in general, fragmented, and are carried out for various purposes. Different population groups are recruited using different protocols and recruitment strategies, making it impossible to compare results or conduct comparable health risk assessments. The recent DEMOCOPHES study was the first attempt to perform harmonized HBM on an European scale, from the study design and protocols to the laboratory measurements and data analysis [1,2]. Furthermore, the ongoing European HBM Initiative HBM4EU is developing harmonized approaches for HBM in Europe, and has recognized trace elements cadmium (Cd), lead (Pb), and mercury (Hg) as priority substances in these HBM activities (https://www.hbm4eu.eu/). Similarly, the Minamata Convention on Mercury was recently ratified, with the aim of protecting human health and the environment from the adverse effects of Hg (www.mercuryconvention.org). Evaluating the effectiveness of this convention includes HBM. In such cases, “*metrology support is of great importance to allow sustainable implementation of the program(s) and provide quality data to assess time and spatial trends*” [3]. The concepts and principles of metrological traceability, which is a prerequisite for obtaining the metrological comparability of measurement results [3], are well elaborated in the *International Vocabulary of Basic and General Terms in Metrology* [4]. However, their practical implementation in various programs has yet to be elaborated.

In summary, two aspects of assuring the appropriate quality of HBM measurement results are critically important: The first is to formulate an appropriate design for protocols and recruitment strategies, and the second is to assess the metrological comparability of the measurement results obtained by participating laboratories. The present paper deals primarily with the second requirement. To demonstrate the equivalence of the measurement results obtained by different laboratories, a case study, including inter-laboratory comparison (ILC) exercises, was implemented in the framework of the EU-funded project PHIME (6th FP, *Public health impact of long-term, low-level mixed element exposure in susceptible population strata*, 2006–2011). Different laboratories in the participating countries within and outside Europe were involved in four ILC exercises, with the aim of determining the total levels of Hg, Cd, and Pb in both freeze-dried and fresh blood human samples.

The overall aim of the PHIME project was to establish associations between low-level exposure to trace elements and critical health effects in susceptible population groups. One of the specific aims was to assess the geographical variability of exposure to such elements in Europe [5]. Within this framework, the present case study was designed to quantify the contribution to the sample size arising from the analytical performance of the methods used. For this purpose, sample sizes were calculated using (1) population variability, (2) measurement uncertainty within a single laboratory, and (3) inter-laboratory variability based on the ILC exercises. Population variability in the blood levels of total Hg, Cd, and Pb was assessed in blood samples from 6–11-year-old children [5]. The collection of samples obtained as part of the PHIME case study and for use in inter-laboratory exercises was conducted in accordance with the Declaration of Helsinki and was approved by the National Medical Ethics Committee of Republic of Slovenia, Ministry of Health, Ljubljana (no. of accordance 98/05/06). All participants and the parents of participating children signed an informed consent form prior taking part in the study.

## 2. Methods

### 2.1. ILC Sample Preparation

Four sets of ILC samples were prepared. Set PT-WB1 included lyophilized human blood from 12 non-exposed persons (each of whom provided 300–400 mL of fresh blood); set PT-WB2 included lyophilized blood from a person occupationally exposed to elemental Hg (this individual provided about 500 mL of fresh blood); and set PT-WB3 included lyophilized human blood from three fish eaters (each of whom provided 250 mL of fresh blood). All samples were collected from voluntary blood donors using the standard blood bag system. The last sample set included three samples of fresh blood from three healthy voluntary donors (samples FF 3613, FF 3614 and GG 0461), collected into several 4 mL lithium heparinized sample tubes (Vacuette, Greiner Bio-One, Kreismünster, Austria).

Blood samples PT-WB1, 2, and 3 were frozen at −20 °C and lyophilized at −47 °C and 0.030 mbar in a freeze dryer (Alpha 1-4, Martin Christ, Osterode am Harz, Germany). Samples were ground in a planetary micro mill (Pulverisette 7, Fritsch Milling and Sizing, Idar-Oberstein, Germany) using agate balls and jars. Particle size was <250 μm. The samples were then sterilized by gamma irradiation at a dose of 5 Mrad using ^60^Co source at the Ruđer Bošković Institute in Croatia to ensure the long-term stability of the material by inhibiting microbial growth. Homogenization was performed by mixing each sample in a rotating plastic container for 48 h. The total weights of the resulting samples were approximately 900 g (PT-WB1), 110 g (PT-WB2), and 150 g (PT-WB3), respectively. Blood samples FF 3613, FF 3614, and GG 0461 were stored at −20 °C before shipping them to the recipient laboratories.

For the purpose of bulk homogeneity testing, 10 aliquots of the PT-WB1 sample and 6 aliquots of the PT-WB2 and PT-WB3 samples, each of ~200 mg, were taken randomly from a rotating plastic container.

Based on the demonstrated homogeneity of the bulk material, the material from the rotating plastic container was then aliquoted into 20 mL glass bottles, each containing 4–5 g of the sample. Between- and within-bottle homogeneity was tested using 6 random bottles of the PT-WB1 sample and 3 bottles each of the samples PT-WB2 and PT-WB3. ILC samples were analyzed at the Jožef Stefan Institute (JSI). Three independent sample aliquots were analyzed. Approximately 500 mg of the sample was dried at 85 °C until it reached a constant weight to determine the dry weight.

In order to guarantee the stability of the samples during the implementation of the ILC, samples from three different bottles per sample set were re-analyzed. The results of the homogeneity and stability tests were obtained using the regular quality system of the JSI laboratory, and are presented in Section 3.1 below.

### 2.2. Analytical Methods Used to Characterize the ILC Samples

To characterize the ILC samples (in terms of homogeneity and stability), the selected elements were determined using: (1) acid digestion and inductively coupled plasma mass spectroscopy; (2) acid digestion and cold vapor atomic absorption spectrometry (CVAAS); and (3) thermal desorption and CVAAS. The measurement procedures are described elsewhere [6,7,8,9,10] and in detail in the following paragraphs. The quality of the described procedures was assured using the reference material Seronorm Whole Blood Level 1 (Sero AS, Billingstad, Norway).

#### 2.2.1. Acid Digestion and Inductively Coupled Plasma Mass Spectroscopy

Approximately 0.2 g of each lyophilized blood sample was weighed into a 50 mL Teflon tube. Six mL 65% HNO_3_ and 2 mL 30% H_2_O_2_ were added and the tube was heated in an aluminum block at 80 °C overnight and then at 130 °C for 2 h. After cooling 2 mL 30% H_2_O_2_ was added two times and the tube was heated each time for 15 min at 90 °C. Finally, the solution was diluted to 40 g with Milli-Q water. Measurements were taken using an Octapole Reaction System Inductively Coupled Plasma Mass Spectrometer (7500ce, Agilent Technologies, Santa Clara, CA, United States) equipped with an ASX-510 Autosampler (Cetac Technologies, Omaha, NE, United States). Helium mode of the Octapole Reaction System (Agilent Technologies) was used. All isotopes were quantified using three central points of the spectral peaks. The instrumental conditions were as follows: nebulizer, Micro Mist; spray chamber, Scott-type; spray chamber temperature, 5 °C; plasma gas flow rate, 15 L/min; carrier gas flow rate, 0.82 L/min; make-up gas flow rate, 0.11 L/min; nebulizer pump, 0.1 rps; RF power, 1500 W; reaction cell gas He 4.1 mL/min; isotopes monitored, ^111^Cd, ^206^Pb, ^207^Pb, and ^208^Pb. The instrument was tuned daily using a solution containing 10 ng/g each of lithium, magnesium, yttrium, calcium, thallium, and cobalt. The detection limit was 0.03 ng/g for Cd and 1 ng/g for Pb.

#### 2.2.2. Acid Digestion and CVAAS

Approximately 0.5 g of blood was weighed in a 50 mL volumetric flask, and after the addition of 1 mL of distilled water, 1 mL of HNO_3_, 1 mL of HClO_4_, and 5 mL of H_2_SO_4_ it was heated at 200 °C on a hotplate for 20 min. After cooling, the digested sample was filled up with distilled water. The measurement of total Hg is based on CVAAS using a semi-automated Hg vapor introduction system. A known volume of the sample was introduced into the reaction vessel. After reduction with SnCl_2_, Hg vapor was swept from the solution by aeration and purged into the absorption cell, where Hg absorbance was measured at 253.7 nm. This method is described in greater detail elsewhere [9]. A standard solution of inorganic mercury (Hg^2+^) prepared from pure elemental Hg (certificate of purity no. 13/2005) was used to calibrate the instrument and check its accuracy. The limit of detection was 0.05 ng/g.

#### 2.2.3. Thermal Desorption and CVAAS

The DMA-80 Direct Mercury Analyzer (Milestone, Shelton, CT, USA) was used. Approximately 0.05 g of sample was weighed in quartz boats and placed in an auto-sampler. The samples were initially dried by an oxygen stream passing through a quartz tube located inside a controlled heating coil. It was then thermally decomposed at 650 °C and the released Hg was reduced on a catalytic column. Hg vapor was collected in a gold amalgamation trap and subsequently desorbed. The Hg content was determined using atomic absorption spectrometry at 254 nm. This method is described in detail elsewhere [11]. A standard solution of inorganic mercury (Hg^2+^) prepared from pure elemental Hg (certificate of purity no. 13/2005) was used to calibrate the instrument and check its accuracy. The limit of detection was 0.08 ng/g.

### 2.3. Measurement Uncertainty

Measurement uncertainty was calculated for the measurement procedures described above. The measurement uncertainty for the total amount of Hg in the blood by thermal desorption and CVAAS was estimated based on the ISO 21748:2010 “Guidance for the use of repeatability, reproducibility and trueness estimates in measurement uncertainty estimation,” using the reproducibility and recovery data from the validation study. The estimated measurement uncertainty for the determination of the total Hg in blood was 6.2% and the expanded uncertainty (k = 2) was 12.5%. The estimation is valid for the “normal” exposure range (i.e., below 5 µg/L).

The measurement uncertainty of Cd and Pb was evaluated using the Nordtest approach from the measurements of the reference materials Seronorm Whole Blood Levels 1 and 2 (Sero AS). The estimated measurement uncertainty for determining the level of Cd in blood was 11.7% (k = 1) at a concentration of approximately 0.3 µg/L (Level 1) and 6.3% (k = 1) at a concentration of approximately 6 µg/L (Level 2). The estimated measurement uncertainty for determining Pb in blood was 10.8% (k = 1) at a concentration of approximately 10 µg/L (Level 1) and 5.1% (k = 1) at a concentration of approximately 310 µg/L (Level 2).

### 2.4. Shipment of Samples

The lyophilized blood samples were shipped in 20 mL bottles at room temperature and frozen fresh blood samples were shipped in 4 mL sampling tubes with cold packs by an accredited courier using high-speed delivery. Deliveries were confirmed by the receiving laboratory.

### 2.5. Analytical Methods Used by the Participating Laboratories

The participants in the ILC exercises were asked to analyze all elements using their validated methodology intended for the HBM case study. They were also asked to make three separate determinations of each element and to report the results, together with a short description of the method used. Using the quality controls applied routinely within the laboratory was a condition of participation. The results for the lyophilized material were reported on a dry weight basis.

The sample pre-treatments and detection procedures used by the participating laboratories are presented in Table 1. The sample pre-treatments included mostly acid digestions (heating or microwave) in various combinations or sole use of perchloric, nitric, or sulfuric acids or hydrogen peroxide; in a minority of cases other oxidizing agents were added. One laboratory used thermal combustion on the samples (DMA). Detection procedures included CVAAS and fluorescence spectrometry (CVAFS), inductively coupled plasma mass spectroscopy (ICPMS), electrothermal atomic absorption spectrometry (ETAAS), flame atomic absorption spectrometry (FAAS), graphite furnace atomic absorption spectrometry (GFAAS), inductively coupled plasma atomic emission spectrometry (ICPAES), and neutron activation analysis (NAA).

All laboratories claimed to have experience in analyzing the selected metals in human blood samples and compliance with the ISO IEC 17025 quality standard; six laboratories were also accredited for trace elements analysis in biological and/or environmental samples. The reference materials used for quality control and assurance and participation in external ILC are listed in Table 1.

### 2.6. Statistical Methods

Between- and within-bottle homogeneity in the ILC exercise was tested using analysis of variance (ANOVA). The coefficient of variation (*C_v_* = standard deviation/mean of measurements) was determined to measure the total Hg in bulk homogeneity testing in the ILC exercise.

Descriptive statistics (mean, standard deviation [SD], geometric mean, geometric SD, CI 95%) of the total Hg, Cd, and Pb concentrations in the blood of children were obtained. The geometric SD was calculated as the SD of log-transformed data.

The sample size required to detect significant differences in the mean concentrations of the selected metals between populations was calculated using STATA 12 software (StataCorp LP, Texas, U.S.). Population means, together with SDs, were used in sample size calculations as an example of population variability. T-tests (one-sample comparison of mean to hypothesized value; H0: Mean _hypothesized_ = Mean _postulated_) were employed to compare population means to postulated means, which differed by 5%, 10%, or 20%, taking into account the SD of the selected population (*SD_pop_*). The statistical powers of 0.80, 0.90, and 0.95 and a significance level α = 0.05 (one-sided) were employed. The predicted ratio of the sample sizes was 1:1.

In order to account for the variability due to measurement uncertainty within one lab and among multiple laboratories, the measurement uncertainty estimated for each element at the JSI lab (*SD_lab_*) and the SD of measurements from the ILC exercises (*SD_inter-lab_*) were used. The combined variability was calculated as:(1)$$SD_{comb1}=\sqrt{SD_{lab}{}^{2}+SD_{pop}{}^{2}}$$
where *SD_lab_* is a measurement of uncertainty (variability within one laboratory) and *SD_pop_* is the standard deviation of values among the selected population; and
(2)$$SD_{comb2}=\sqrt{SD_{inter-lab}{}^{2}+SD_{lab}^{2}+SD_{pop}{}^{2}}$$
where *SD_inter-lab_* is the standard deviation of measurements among different laboratories (inter-laboratory variability).

## 3. Results

### 3.1. The Homogeneity and Stability of the ILC Samples

According to the results of the bulk homogeneity testing, which yielded coefficients of variation of 7.3%, 0.9%, and 0.8% for total Hg in PT-WB1, 2, and 3, respectively, the bulk materials were homogenous. Between- (s_bb_) and within-bottle (s_wb_) homogeneity testing revealed no statistically significant variability. The maximum within-bottle homogeneity was 12%, while the between-bottle homogeneity was 4.5%; both were observed in the case of Pb. The homogeneities for Hg and Cd were lower (within-bottle < 7%, between-bottle < 2%). These values indicate that the materials were confirmed to be homogenous for all tested elements and were therefore suitable for distribution to laboratories. Stability testing showed no significant differences between the three testing periods within 6 months and confirmed the stability of selected elements for the duration of the ILC exercises. The residual moisture content (expressed as the mean ± SD, *n* = 4–6) was 3.4 ± 0.4%, 4.4 ± 0.2%, and 4.8 ± 0.2% for PT-WB1, 2, and 3, respectively.

### 3.2. Total Hg, Cd and Pb in Blood: Results of the ILC Exercise

Among the three selected metals, the measurement variability observed between the participating laboratories was the highest in the case of Cd (Figure 1). The fresh blood samples, representing a concentration range considered normal for the general adult population (means of 0.54, 0.46, and 0.48 ng/g), showed somewhat higher variability in the Cd concentrations (26%, 34%, and 31%, respectively) compared to the lyophilized samples with respective mean concentrations of 3.3 (27%), 3.1 (27%), and 9.6 ng/g (18%) (Figure 1). One laboratory reported inadequate levels in PT-WB1 (11.3 ng/g), which was therefore excluded from the total mean and SD calculation.

Total Hg measurements showed inter-laboratory variabilities of 12% (mean 8 ng/g), 15% (mean 54 ng/g), and 16% (mean 103 ng/g) in the lyophilized material (Figure 1), and, similarly, between 9% and 18% in the fresh blood samples at levels that are typical for people who consume fish three or more times a month (means 1.09, 1.39, and 1.80 ng/g, respectively). In the latter, the variability depended on the concentration level, the highest being in the lowest concentration range. Three laboratories reported inadequate levels for PT-WB1, with inadequate limits of detection (199 ng/g, <50 ng/g, <50 ng/g). These results were excluded from the total mean and SD calculations.

The respective inter-laboratory variabilities for Pb measurements were 10% (mean 159 ng/g), 16% (mean 158 ng/g), and 20% (mean 71 ng/g) for lyophilized material (Figure 1) and between 4% and 22% in the fresh blood samples, the latter representing concentrations found in non-exposed populations (mean levels between 25 and 31 ng/g). Four laboratories reported inadequate levels in the PT-WB1 (55, 315, 573, and 781 ng/g, respectively), which were therefore excluded from the total mean and SD calculations.

### 3.3. Total Hg, Cd and Pb in Blood: Population Variability

Population variability was estimated based on the concentrations of total Hg, Cd, and Pb measured in the blood of children from Slovenia, a case study implemented in the framework of the PHIME project. The Slovenian study population consisted of residents of urban, rural, and Hg-contaminated (the town of Idrija) areas. The results of the urban population group were presented along with other participating regions by Hrubá et al. [5]; however, for the purposes of the present study, children from the rural and contaminated areas were also included. The Slovenian study population relied on a population group of 6- to 11-year-old schoolchildren, among whom both genders and all age groups were equally represented. The participating children came from different socioeconomic backgrounds. All children were recruited, sampled, and analyzed simultaneously, following the same protocol described in Hrubá et al. [5]. The descriptive statistics for the Hg, Cd, and Pb levels in the blood of the Slovenian study population are presented in Table 2.

The international variability reported by the authors [5] was considerable for blood Hg, and varied by a factor of 8 between European cities, with the highest levels observed in the Slovenian children. Cd and Pb were only marginally different between the European cities, with the highest geometric mean being 1.5 times that of the lowest [5]. Within the Slovenian study population, Hg blood levels were lower in the rural population than in the urban group (*p* = 0.011), while no significant difference was observed between the urban and Hg-contaminated areas (*p* = 0.355). However, Cd levels were higher in the rural and contaminated area populations than in the urban study group (both *p* < 0.001); the same trend was observed for Pb (*p* = 0.055 and 0.035, respectively) (Table 2).

For the purpose of the power size calculations presented in this paper, the levels in the urban population were considered, as this particular area type was used in the international comparison conducted by Hrubá et al. [5]. All three variables were distributed log-normally, so geometric means and geometric SDs were considered in the sample size calculations.

### 3.4. Sample Size Calculation

Based on the observed variability within the urban study population group, the sample size required to observe a significant difference in blood Hg level was calculated and is presented in Figure 2. For this calculation it was assumed that all measurements were performed in one laboratory using the CVAAS method (described in Section 2.2). In order to observe a 20% difference in a mean Hg level between two populations or population groups, a minimum of 71 participants is required at statistical power of 0.90 and a significance level of 0.05. Lower detected differences in mean levels require a considerably higher number of participants—e.g., 283 participants are required to observe a 10% difference and 1131 participants are required to observe a 5% difference (Figure 2).

The population variability observed in evaluating the blood Cd levels in urban Slovenian children necessitated a sample size of 18 participants to reveal 20% difference in means between two populations or population groups, assuming that all measurements are performed in one laboratory (described in Section 2.2). Seventy participants would be required to reveal a significant difference of 10% and 280 participants would be required to reveal a 5% difference, with statistical power of 0.90 and a significance level of 0.05 (Figure 3).

The population variability observed in evaluating blood Pb in urban Slovenian children necessitated a sample size of 27 participants to reveal a 20% difference in means between two populations or population groups, assuming that all measurements were performed in one laboratory (described in Section 2.2). To reveal significant differences of 10% and 5%, the sample size would have to be increased to 105 and 418 participants, respectively, taking into account a statistical power of 0.90 and significance level of 0.05 (Figure 4).

#### 3.4.1. Measurement Uncertainty within a Single Laboratory

Measurement uncertainty at relevant concentration ranges was calculated to account for analytical variability within a single laboratory (*SD_lab_*). Considering the measurement uncertainty calculated for the analytical procedures within the JSI lab (Section 2.4), the required sample sizes would need to be increased from 283 to 286 (Hg), 70 to 81 (Cd), and 105 to 115 (Pb) (Table 3).

#### 3.4.2. Measurement Variability among Multiple Laboratories

Accounting for the inter-laboratory variability (*SD_inter-lab_*) observed in the ILC exercise (Figure 1), the required sample sizes would need to be increased by an additional 5–10% in the case of Hg, 75–220% in the case of Cd, and 1–37% in the case of Pb blood measurements (Table 3). The range represents minimal to maximal increases in the sample sizes, depending on the ILC sample set used. Only the sample sets of the relevant concentration range were considered in the calculations.

## 4. Discussion

With the aim of evaluating the significance of measurement variability among multiple laboratories, relative to the variability within a single laboratory and to the study population, the prerequisite in this study was to prepare a sufficient quantity of uniform sample material for distribution among the different laboratories. The material was prepared from fresh blood samples collected from healthy volunteers. Lyophilization and homogenization of the collected blood samples provided homogenous materials for all tested elements, which were suitable for distribution to the participating laboratories, and were confirmed to be stable for the duration of the ILC exercises. Because lyophilized samples contained higher concentrations of elements than fresh blood samples, they were not representative of routine samples. The presented ILC exercise therefore also included a limited number of fresh blood samples with the purpose of validating the variability of the relevant concentration ranges. They showed concentration ranges of the studied metals that are typically observed in general population groups within Europe. Although the inter-laboratory variability of the fresh blood samples was somewhat higher than in the lyophilized ILC samples, they did not differ considerably. The exposure levels observed in the study population that were used to assess population variability were more or less similar to other studies that have reported on the aforementioned elements in children’s blood. Certain (although not marked) differences include undetected Cd levels in the German and US surveys [12,13] and somewhat higher Pb levels identified in the Czech HBM study [14] while lower ones were found in the US [12]. Compared to the Slovenian study population, Hg levels were lower in the German, US, and Czech studies [12,14,15], which is in line with the international comparison conducted by Hrubá et al. [5], which found that Slovenian children had the highest Hg levels among the participating European countries. Overall, the levels observed in our study population were within exposure levels that do not present an elevated health risk according to the reference values established for the European population [14,16].

Based on the variability within the study population, the highest required sample size was calculated for Hg measurements. The latter had the highest population variability (57%) among the three metals studied. Lead and Cd in the blood showed variabilities of 35% and 29%, respectively. Accordingly, the lowest required sample size was observed for Cd (Figure 2, Figure 3 and Figure 4).

The sample size calculation results showed a negligible increase (1%) in the required sample size if the measurement uncertainty within one laboratory was in the range of 10% of the population variability (Hg), while an increase of 10% was observed in the case of Pb with an uncertainty of one-third of the population variability. The highest increase occurred in the case of Cd (16%), with its measurement uncertainty representing 40% of the population variability. If the hypothetical uncertainty was equal to population variability, the required sample size would double.

If the measurement variability among multiple laboratories participating in the ILC was added, the increase in the required sample size would be considerable, particularly in the case of Cd (Table 3, Figure 5). Cadmium blood levels, their low variability among the urban study group, and high inter-laboratory variability reflect the great significance of centralized analysis. However, as observed in the case of blood Hg measurements, high population variability decreased the influence of multiple laboratory analyses.

The calculated sample size is in line with the Slovenian case study, including three different population groups (urban, rural, and Hg-contaminated; see Table 2). The observed differences in blood Hg concentrations between rural and urban (25%) and between rural and contaminated areas (23%) were in the range of the calculated sample size requirements and was shown to be significant, while that between urban and Hg-contaminated was not (2% difference in means). The sample size used in the study of children [5], with the number of participating children being between 21 and 57, depending on the country, was also sufficient to observe significant differences between the participating European countries, with the means ranging from 0.21 µg/L in Poland to 0.94 µg/L in Slovenia. In this study Hg blood levels were determined by two laboratories, which showed excellent agreement [5].

The between-group difference of more than 60% in mean blood Cd levels in the Slovenian case study was clearly significant (Table 2). The international levels reported by Hrubá et al. [5] showed very little variation in blood Cd between the European countries, but due to the centralized laboratory analysis, the differences observed between the participating cities were statistically significant.

In the case of blood Pb, the difference of 20% between urban and rural populations was revealed to be only marginally significant, which was due to the variability within the rural study group being almost double that of the urban group used in the sample size calculations. In this particular case we would need 118 participants in each group. Consistent with the calculations presented in this paper, the difference of 22% observed between urban and industrial population was revealed to be statistically significant.

From the sample size calculations using three commonly studied metals, it is obvious that requirements for sample size in HBM studies vary considerably. They depend on the selected analyte(s) and its/their variability within study groups, as well as the number of laboratories involved in sample analysis and the degree to which they agree with the reported results. As the HBM studies usually aim to measure different toxicants or their metabolites in the same subjects, the sample size requirements should be based on the one with the highest population variability, which in our case was Hg. Depending on the desired effect size to be observed, sample size increases following a square function. The effect size a particular study aims to observe is a matter of exposure range in a certain population group and toxicological relevance. In the case study described, an effect size with a minimum 20% difference was observed to be statistically significant for all selected elements and seemed relevant from the exposure perspective. Such an effect size requires at least 79 participants in each study group, taking into account inter-laboratory variability. A difference of 10% increased the sample size to approximately 300; a further reduction in effect size to 5% requires a sample size exceeding 1200 participants, which is regularly achieved in national HBM surveys such as those conducted in Germany, France, the Czech Republic, Slovenia, the United States, etc. [12,14,17,18]. Although health effects could occur at relatively low levels of exposure, as has been debated for methyl Hg in children [19], the toxicological relevance of such a small effect size is still to be clarified for many of the environmental pollutants, with an emphasis on susceptible population groups. Nevertheless, its importance in time-trend studies and exposure pathway identification, as the main objectives in HBM, remains. In spite of the low levels observed in children living in the selected European countries, strict analytical quality control and centralized analyses conducted by Hrubá et al. [5] revealed an association of blood Hg with amalgam fillings and fish intake; it was also found that blood Cd was associated with living close to dense traffic. Small changes can be especially relevant in assessing micro-elements (e.g., Zn, Cu, Se) with a rather narrow concentration range.

The present study has certain limitations, including the type of the ILC samples used (lyophilized blood), the corresponding concentration ranges, and the number of laboratories analyzing the ILC samples, which differed between the elements selected for the study. However, the measurement variability obtained from the analysis of lyophilized blood was validated by the analysis of fresh blood containing the relevant concentration ranges. Moreover, for each element, measurement variability within a single laboratory was compared to the variability among multiple laboratories and their relevant contributions to the required sample sized were assessed.

Despite the limitations, the data collected in this study clearly demonstrate that analyses of biological samples in an HBM study should preferably be performed in a single laboratory (i.e., a centralized analysis should be done) in order to minimize measurement variability. Laboratories must assure quality control using a regular quality system and participate in ILC exercises to achieve a relevant concentration range and matrices. The laboratory should also prove adequate measurement uncertainty in the range, as demonstrated in our case study. Uncertainty that constitutes 10% of the population variability contributes negligibly to the required samples size, while uncertainty constituting one-third of the population variability increases sample size requirements for 10%.

However, the choice of a single laboratory is not feasible in a practical sense, as “cross-European” and other international studies are usually beyond the capacity of one laboratory. This increases the importance of regular and strict quality controls, including the use of appropriate certified reference materials (CRMs) and the participation in international laboratory comparisons. This is where the support of national metrology institutes and regional metrology organizations, such as the European Association of National Metrology Institutes, in quality control procedures is required [20]. The development of harmonized approaches for HBM, such as the recently implemented COPHES (Consortium to Perform Human biomonitoring on a European Scale) project and the currently ongoing European HBM Initiative (HBM4EU) should also provide the means for sustainable metrology support in order to guarantee the comparability of the results over time and throughout geographical regions, be that Europe or worldwide.

Metrological comparability is required not only for measurements of chemical substances, but must extend to the application of new biomarkers that can be applied in molecular epidemiology, genetic toxicology, “omics-technology”, epigenetics, and bioinformatics. These new biomarkers are crucial to the identification of toxicity pathways in humans, and will provide new insights and applications in HBM and environmental health studies. Metrology support will strengthen these initiatives with improved quality, reliability, and comparability of measurements to ensure and strengthen the position of HBM at the European level.

## 5. Conclusions

The present work evaluated the significance of measurement variability within a single laboratory or among multiple laboratories and its influence on HBM design, particularly sample size, compared to population variability. Measurement uncertainty within one laboratory for Hg, Cd, and Pb in blood showed little influence on the sample size requirements, while the inter-laboratory variability of the three metals increased the requirements considerably, particularly in the case of low population variability, as was observed for Cd in blood. Such low variability is common in general populations, where no contamination sites (“hot spots”) are present but the potency of the substances analyzed are still within the range capable of causing health effects in susceptible population groups. In such cases, the effect of the measurement variability can be reduced using a single laboratory approach (centralized analysis). The approach used in this study should also be applied in planning proficiency testing schemes where z-scores are used to measure the performance of a particular laboratory. The z-scores require predefined quality standards, which should be carefully defined prior to the proficiency study, and should take into account the objectives of the HBM in terms of time and spatial trends.

## Figures and Tables

**Figure 1 ijerph-16-02287-f001:**
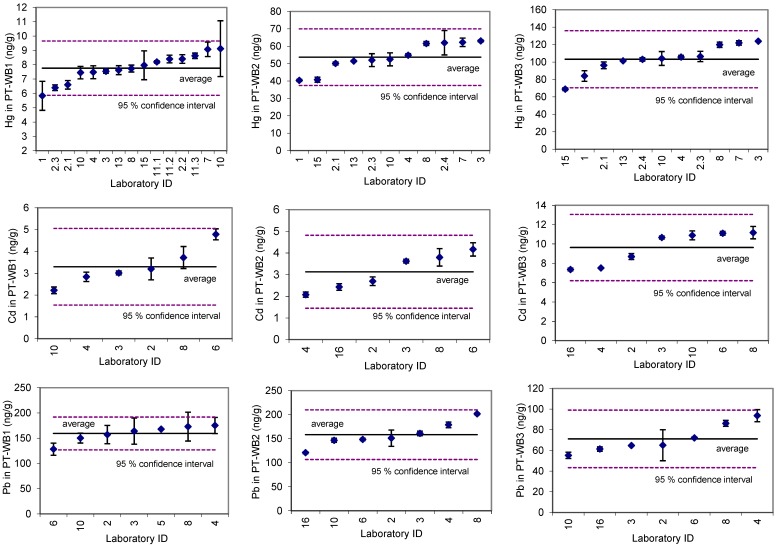
Total Hg, Cd, and Pb concentrations in the lyophilized ILC samples from the participating laboratories.

**Figure 2 ijerph-16-02287-f002:**
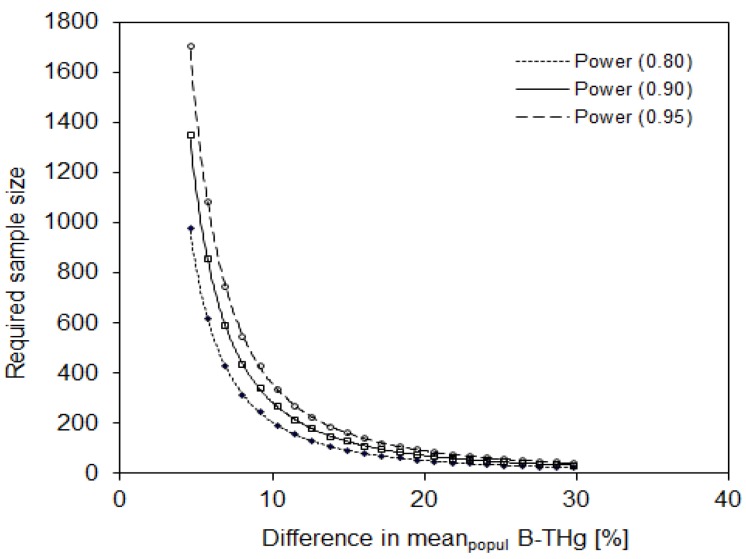
The sample size required to reveal a significant difference in blood Hg levels between two population groups, assuming that all analyses are performed in one laboratory. Statistical power = 0.80, 0.90, and 0.95; α = 0.05; one-sided.

**Figure 3 ijerph-16-02287-f003:**
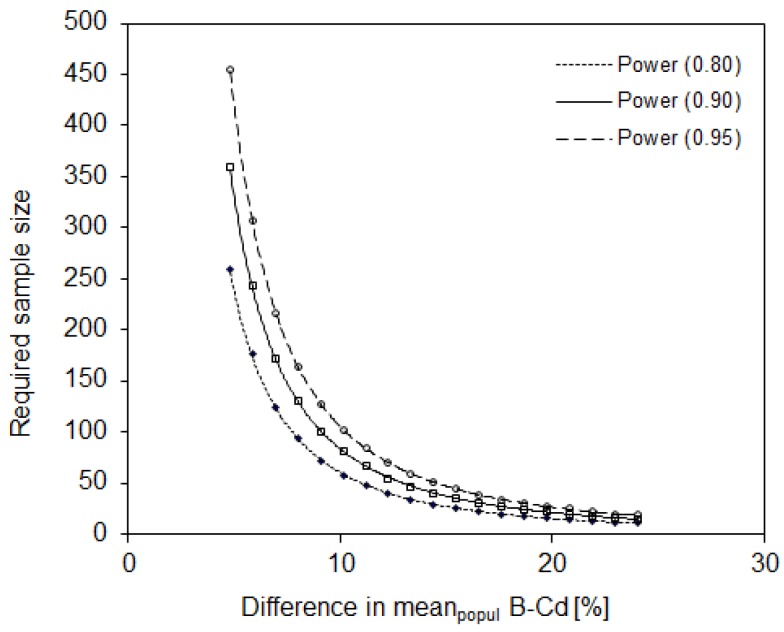
The sample size required to reveal a significant difference in mean blood Cd between two population groups, assuming that all analyses are performed in one laboratory. Statistical power = 0.80, 0.90, and 0.95; α = 0.05; one-sided.

**Figure 4 ijerph-16-02287-f004:**
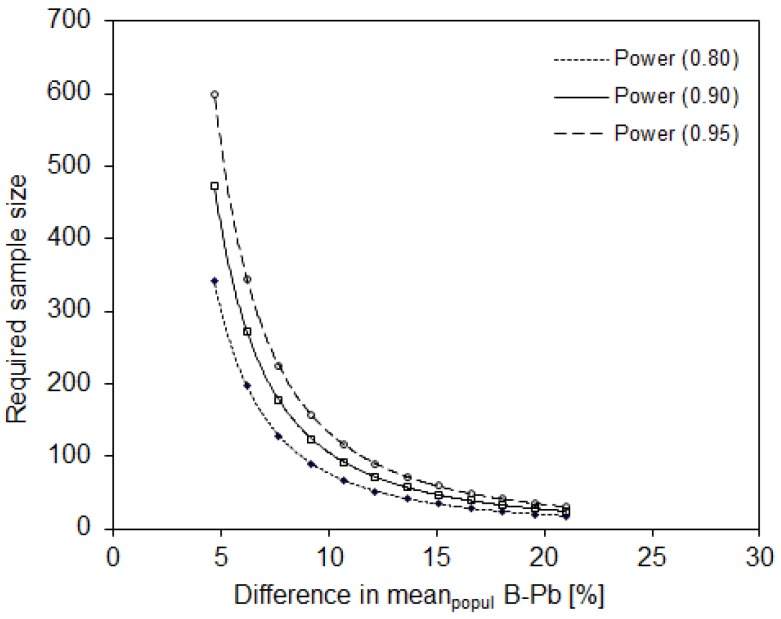
The sample size required to reveal a significant difference in mean blood Pb between two population groups, assuming that all analyses are performed in one laboratory. Statistical power = 0.80, 0.90 and 0.95; α = 0.05; one-sided.

**Figure 5 ijerph-16-02287-f005:**
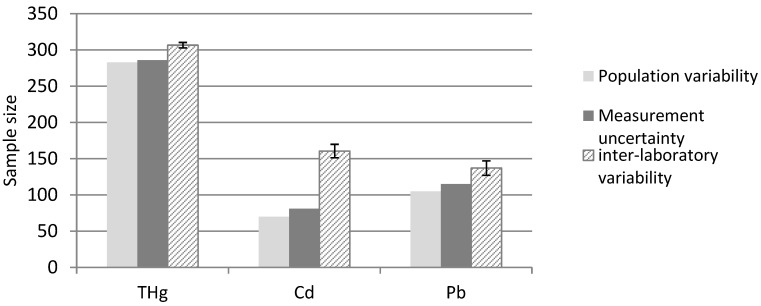
The sample size required to detect a 10% difference in mean values between two population groups depending on the (1) population variability, (2) population variability and measurement uncertainty within one lab, and (3) variability among different laboratories (inter-laboratory variability). Bars represent variation due to the use of different ILC samples. ILC, inter-laboratory comparison.

**Table 1 ijerph-16-02287-t001:** List of sample pre-treatments and measurement methods used by the participating laboratories. ILC, inter-laboratory comparison.

Lab ID	Pre-Treatment (Acids/Alkalines Used, T and Duration of Digestion)	Reported Analytes/Detection			Quality Control (Reference Materials and ILC)	Accreditation
		Hg	Cd	Pb		
1	HClO_4_/HNO_3_ (5:1), 70 °C, 16 h, open-lose caps	CVAFS	ICP MS	ICP MS	no data	
2	HNO_3_/H_2_O_2_, 120–130 °C in closed vessels; none for DMA	CVAAS, DMA	ICPMS	ICP MS	Seronorm WB1, NIST 1547, NIST 966, IAEA-086	no
3	Ultraclave, 250 °C, 1 h	ICPMS	ICPMS	ICPMS	CRM NIST 966, Seronorm WB1	no
4	Pb, Cd: HNO_3_/H_2_O_2_, Milestone Mega1200+FAM 40, 20 min; Hg: none	CVAAS	ETAAS (Zeeman)	ETAAS (Zeeman)	Seronorm WB1	yes
5	HNO_3_/H_2_O_2_ in closed vessels, Milestone MLS 1200 MEGA	-	-	ICPMS	Dorm2, Dolt3, BCR 184, NIST 1577b	yes
6	HNO_3_/H_2_O_2_ in closed vessels, FAR IR elect. oven, <140 °C, 4 h	-	ETAAS (Zeeman)	ETAAS (Zeeman)	ClinChek (whole blood), internal reference material	
7	HNO_3_ in closed vessels, A.Parr Multiwave	CVAAS	-	-	Seronorm WB1; WB2, WB3	no
8	Hg: HNO_3_, 100 °C 4 h in closed teflon vessel, Pb, Cd: deproteinization after dissolution in water	CVAAS	ETAAS (Zeeman)	ETAAS (Zeeman)	BCR 194, 195; ILC: CDC, USA; Instituto Superiore di Sanita, Italy	no
9	Hg: HNO_3_/H_2_SO_4_; Pb and Cd: 10 × dilution with solution containing Triton X-100, EDTA, ammonia	CVAAS	ICPAES	ICPAES	no data	yes
10	HNO_3_/H_2_O_2_, 160 °C, 10 h; closed vessels	ICPMS	ICPMS	ICPMS	Dorm-2, Dolt-3, ClinCheck (serum)	no
11	Three procedures were used: (1) HNO_3_, 95 °C, 3 h; (2) Aqua regia, 95 °C, 30 min; (3) KOH/BrCl, 95 °C, 3 h	CVAFS	-	-	SRM 1566b, Dorm-2	yes
12	HNO_3_/HClO_4_ in open vessels, 130 °C, 4–5 h	NAA	FAAS Deuterium	FAAS Deuterium	IAEA A-13	yes
13	HNO_3_/H_2_SO_4_/HClO_4_ (1:5:1) in open flasks, 30 min, 200–230 °C	CVAAS	-	-	no data	
14	HNO_3_ in closed vessels, Microwave oven, 160 °C 15 min for Hg, 190 °C 10 min for Pb, Zn	CVAAS Deuterium	-	GFAAS Deuterium	no data	yes
15	dilution in H_2_O, Magos method	CVAAS	-	-	Seronorm WB1; ILC: International comparison program Centre de toxicoiogie du Quebec	no

**Table 2 ijerph-16-02287-t002:** Population variability: total levels of Hg, Cd, and Pb in the blood (µg/L) of 6- to 11-year-old children from three study areas in Slovenia. GM, geometric mean; Mean, arithmetic mean; min-max, range; *n*, sample size; 95% CI, 95% confidence intervals; SD, standard deviation.

Analyte Study Area	*n*	Min-Max	Mean (SD)	GM (SD)	95% CI
**Total Hg**					
All	174	0.35–4.39	0.97 (0.64)	0.84 (0.55)	0.78–0.91
Urban	45	0.35–3.05	1.05 (0.53)	0.94 (0.54)	0.82–1.08
Rural	66	0.35–3.72	0.82 (0.57)	0.71 (0.44)	0.63–0.80
Hg-contaminated	63	0.41–4.39	1.08 (0.75)	0.92 (0.63)	0.81–1.05
**Cd**					
All	150	0.13–0.69	0.23 (0.12)	0.20 (0.12)	0.19–0.22
Urban	42	0.09–0.28	0.15 (0.04)	0.14 (0.04)	0.13–0.16
Rural	65	0.13–0.54	0.25 (0.11)	0.23 (0.12)	0.20–0.25
Hg-contaminated	43	0.13–0.69	0.27 (0.14)	0.24 (0.15)	0.20–0.28
**Pb**					
All	165	5.33–56.8	16.7 (7.27)	15.4 (7.15)	14.6–16.4
Urban	42	6.90–23.7	14.0 (4.08)	13.4 (4.68)	12.3–14.8
Rural	64	5.33–56.8	17.8 (9.29)	16.1 (8.84)	14.4–16.5
Hg-contaminated	59	7.38–36.9	17.3 (6.08)	16.3 (6.81)	14.9–17.9

**Table 3 ijerph-16-02287-t003:** The sample size (*n*) required to observe 5, 10, and 20% differences in blood levels of Hg, Cd, and Pb between population groups (effect size). Combined SDs include *SD_lab_*, *SD_inter-lab_*, and *SD_pop_*. Statistical power = 0.90; α = 0.05; one-sided. GM, geometric mean of the study population; SD, standard deviation.

				*n* According to the Effect Size					*n* According to the Effect Size		
	GM (SD)	*SD_lab_*	*SD_pop_* + *SD_lab_*	5%	10%	20%	*SD_inter-lab_*	*SD_pop_* + *SD_lab_ + SD_inter-lab_*	5%	10%	20%
**Hg**	0.94 (0.54)	0.06	0.543	1144	286	72	0.113–0.170	0.555–0.569	1195–1256	299–314	75–79
**Cd**	0.14 (0.04)	0.02	0.043	324	81	21	0.036–0.048	0.057–0.064	568–716	142–179	36–45
**Pb**	13.4 (4.68)	2.95	4.91	460	115	18	0.536–2.95	4.94–5.72	466–625	117–157	30–40

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
