# Peer review of "Analytical Quality Requirements in Human Biomonitoring Programs: Trace Elements in Human Blood"

_ijerph, 2019, doi:10.3390/ijerph16132287_

Round 1

Reviewer 1 Report

The manuscript discusses the significance of measurement variability within a single laboratory or among multiple laboratories that influence HBM design, particularly sample size, as compared to the population variability. It is unclear that the number of samples tested for metals are large enough to reach to the conclusion of the authors. Needs to include the limitation of the works that possibly relate to the sets and number of samples considered in the laboratories. The manuscript is not easy to follow and requires restructuring and improving written English.

Author Response

The manuscript discusses the significance of measurement variability within a single laboratory or among multiple laboratories that influence HBM design, particularly sample size, as compared to the population variability. It is unclear that the number of samples tested for metals are large enough to reach to the conclusion of the authors. Needs to include the limitation of the works that possibly relate to the sets and number of samples considered in the laboratories. The manuscript is not easy to follow and requires restructuring and improving written English.

Response: We thank the reviewer for critical evaluation of the manuscript. We have added limitations of the study to the Discussion and re-structured some parts, particularly the Discussion, to make it easier to follow.The manuscript has been revised also by a professional English editing service. However, the reviewer reported that »The quality of the writing and the organization of the paper were already very strong, so there were few outright errors to correct.«

Reviewer 2 Report

Current recommendations in HBM studies do not always describe tests in a precise way and do not include all issues important to assure reliable results. Also, the manner of conducting tests is not in full agreement with the acceptance criteria. Thus, the reliability of bioanalytical methods is an important  topic of interest.

I have the following doubts and questions as to the correctness of the described tests, possibly as to the correct way of writing about the methodology:

1)      On page 2 , lines 75-79 authors wrote:

„Sample set PT-WB1 included 12 samples of blood (300-400 mL each);

sample  set  PT-WB2  consisted  of  about  500  mL  of  blood  that  was  collected  from  a  person (!) occupationally exposed to elemental Hg; sample set PT-WB3 included three samples of blood (250 mL each) that were collected from three fishermen…”

-          does it really mean that each single person gave 300 to 400 mL or even a half of liter of blood for the presented studies??? Which modern sensitive analytical technique REQUIRES a half of liter of blood taken from a person???  Does it mean 1 sample of blood of non-exposed person= 400mL? 12x 400mL= 4.8 L for a set PT-WB1?

2)      Page 3, line 96 : “Bottles were analysed at the Jožef Stefan Institute (JSI).” Authors analyzed SAMPLES not bottles.  

3)      Page 3, line 100 : “…a set of three bottles was re-analysed”- the same error as mentioned above

4)      Page 3, line 104 : “analysis of selected elements was done”- We CANNOT ANALYSE elements, we determine them by ICP-MS or using a different technique

5)      Page 3, line 132:  The author of a method is given (Akagi, 1997).  Authors should include the number as in  the references [9].

6)      Why the authors adapted a methodology form 1997 (22 years ago) for current Hg determination  (CVAAS)? Does it mean that a faster, cheaper, and more precise analytical procedures (even required also for CVAAS measurements) does not exist at present?

7)   what was a standard solution for the analysis of total mercury in the sample? A solution of inorganic mercury (II)? It was not given in the text.

8)      For people who eat large quantities of fish and shellfish, most mercury contained in the red blood cells is in the form of methylmercury; therefore, the methylmercury exposure can be evaluated by measuring total mercury in blood. The exposure to inorganic mercury/mercury vapor can be evaluated by measuring the total mercury in plasma. Did the authors take it into account? No information about speciation studies ( even in general) is provided.

9)      I suggest to include a column with references  in Table 1. Please add LOD and LOQ values in the same table

10)  In Table 1, LAB ID:11- rewrite the pre-treatment procedures to make it more clear for readers

11)  No limitations of the study were mentioned. An important aspect of variability - temporal biological variability - was not assessed, as participants were not recalled to repeat measurements on another day. Thus, for a thorough evaluation of the measurement reliability, consideration of this within-participant day-to-day variability would require incorporation of data from other studies.

12)  The number of reanalyzed clinical samples is often questioned in similar studies.  It is a good subject of discussion in the manuscript.

13)  Please, specify: in the PHIME project (page 2) which ended in 2011 - what was done exactly and what is a new achievement in the presented manuscript? Do I understand correctly that since "objective of the present case study was to quantify the contribution to the sample size arising from analytical performance of the methods used” –line 60-  this goal was not included in the project completed in 2011?

14)   Page 9, line 270: “…..is presented in 0” – ?

Author Response

Current recommendations in HBM studies do not always describe tests in a precise way and do not include all issues important to assure reliable results. Also, the manner of conducting tests is not in full agreement with the acceptance criteria. Thus, the reliability of bioanalytical methods is an important  topic of interest.

I have the following doubts and questions as to the correctness of the described tests, possibly as to the correct way of writing about the methodology:

1)      On page 2 , lines 75-79 authors wrote:

„Sample set PT-WB1 included 12 samples of blood (300-400 mL each);

sample  set  PT-WB2  consisted  of  about  500  mL  of  blood  that  was  collected  from  a  person (!) occupationally exposed to elemental Hg; sample set PT-WB3 included three samples of blood (250 mL each) that were collected from three fishermen…”

-          does it really mean that each single person gave 300 to 400 mL or even a half of liter of blood for the presented studies??? Which modern sensitive analytical technique REQUIRES a half of liter of blood taken from a person???  Does it mean 1 sample of blood of non-exposed person= 400mL? 12x 400mL= 4.8 L for a set PT-WB1?

Response: Correct. Each person provided 300-500 mL of blood which allowed us to prepare sufficient quantity of lyophilized blood for inter-laboratory comparison exercise. The text in the Section 2.1 has been re-written to make it more clear.

2)      Page 3, line 96 : “Bottles were analysed at the Jožef Stefan Institute (JSI).” Authors analyzed SAMPLES not bottles.

Response: corrected

3)      Page 3, line 100 : “…a set of three bottles was re-analysed”- the same error as mentioned above

Response: corrected

4)      Page 3, line 104 : “analysis of selected elements was done”- We CANNOT ANALYSE elements, we determine them by ICP-MS or using a different technique

Response: corrected

5)      Page 3, line 132:  The author of a method is given (Akagi, 1997).  Authors should include the number as in  the references [9].

Response: corrected.

6)      Why the authors adapted a methodology form 1997 (22 years ago) for current Hg determination  (CVAAS)? Does it mean that a faster, cheaper, and more precise analytical procedures (even required also for CVAAS measurements) does not exist at present?

Response: The method by Akagi (1997) is one of the most commonly used methods for determination of total Hg in biological samples, such as blood and urine. It is cheap and fast in compare to other CVAAS or CVAFS methods available. The precision is equivalent to other CVAAS methods. However, lower detection limit (but not necessarily higher precision) can be achieved using CVAFS, but the latter is more expensive and time consuming and was not required for the type of samples used in the study and the concentration ranges observed. All four sets of inter-comparison samples were analyzed in parallel also with the Direct Mercury Analyzer (DMA-80), which showed equivalent results.

7)   what was a standard solution for the analysis of total mercury in the sample? A solution of inorganic mercury (II)? It was not given in the text.

Response: Yes, a solution of Hg2+ prepared from pure elemental Hg (certificate of purity no. 13/2005) was used for calibration. At the time of this study, a solution traceable to NIST was not available. We have added specification of the solution used to the text in the Section 2.2.

8)      For people who eat large quantities of fish and shellfish, most mercury contained in the red blood cells is in the form of methylmercury; therefore, the methylmercury exposure can be evaluated by measuring total mercury in blood. The exposure to inorganic mercury/mercury vapor can be evaluated by measuring the total mercury in plasma. Did the authors take it into account? No information about speciation studies ( even in general) is provided.

Response: The purpose of the paper was to assess analytical performance/requirements for total Hg in whole blood, which is the most commonly used biomarker in HBM and other population-based studies. Speciation of Hg was actually performed in few of the participating laboratories, but we had insufficient data to evaluate variability as we did for total Hg.

9)      I suggest to include a column with references  in Table 1. Please add LOD and LOQ values in the same table

Response: Unfortunately, we cannot include references to the methods in Table 1, because this information was not provided by the participating laboratories. Also the LOD/LOQ values were not reported by all laboratories. However, lack of LOD/LOQ information does not limit the study, as the concentration  ranges of the inter-laboratory comparison samples were considerably above the LOD/LOQ. The results of the laboratories with inadequate LODs were excluded from statistical analysis.

10)  In Table 1, LAB ID:11- rewrite the pre-treatment procedures to make it more clear for readers

Response: we have adapted the text to make it more clear. The LAB 11 used 4 different procedures and delivered 4 separate results. The results of the 4th procedure (combustion) were not taken into account as they deviated from the others and this procedure was removed.

11)  No limitations of the study were mentioned. An important aspect of variability - temporal biological variability - was not assessed, as participants were not recalled to repeat measurements on another day. Thus, for a thorough evaluation of the measurement reliability, consideration of this within-participant day-to-day variability would require incorporation of data from other studies.

Response: We have added limitations of the study to the Discussion. However, assessment of temporal biological variability was out of the scope of the present study. We demonstrated significance of measurement variability for observing spatial or temporal differences on a population level. In order to evaluate measurement variability within one participant, repeated individual measurement would be required and the relative differences obtained should be compared to the measurement uncertainty of the method used. The latter should be lower than the inter-individual difference observed to claim it significant.

12)  The number of reanalyzed clinical samples is often questioned in similar studies.  It is a good subject of discussion in the manuscript.

13)  Please, specify: in the PHIME project (page 2) which ended in 2011 - what was done exactly and what is a new achievement in the presented manuscript? Do I understand correctly that since "objective of the present case study was to quantify the contribution to the sample size arising from analytical performance of the methods used” –line 60-  this goal was not included in the project completed in 2011?

Response: The goal of this paper was not one of the main objectives of the PHIME project, which dealt with critical health outcomes in relation to chronic low level exposure to various environmental pollutants, and included many case studies. However, one of the specific objectives within the PHIME was also to evaluate inter-laboratory variability for the three metals, because the children study, which compared exposure between different European countries, was one of the case studies within the PHIME (Hruba et al., 2012). We have adapted the text on page 2 accordingly.

14)   Page 9, line 270: “…..is presented in 0” – ?

Response: Cross-referencing of the tables and figures somehow collapsed. We have manually entered the corresponding references to the Tables and Figures.

Round 2

Reviewer 2 Report

I have doubts whether such a large volume of blood for research was really necessary.